# Is Time’s Asymmetry Related to Irreversible Processes and the Second Law?

**DOI:** 10.3390/e25091297

**Published:** 2023-09-05

**Authors:** Arieh Ben-Naim

**Affiliations:** Department of Physical Chemistry, Edmond J. Safra Campus, The Hebrew University, Jerusalem 91904, Israel; ariehbennaim@gmail.com

**Keywords:** entropy, time, Second Law, thermodynamics, irreversible processes

## Abstract

In this article, we start by describing one of the most characteristic properties of time: “*time can never decrease*”. From this property, numerous authors have concluded that irreversible processes, that *always* proceed in one direction, must be related to time’s arrow. It is shown that while time’s decrease can never occur, irreversible processes can be reversed, although with extremely low probability. Similarly, it is argued that both entropy and the Second Law are *timeless*, i.e., have nothing to do with either time or with time’s arrow.

## 1. Introduction

The association of entropy and the Second Law with time is quite old and goes back to Clausius, who coined the word entropy and formulated the Second Law in terms of the ever-increasing entropy of the universe. More explicitly, Clausius made the following statement of the Second Law:


*
**“Entropy of the Universe always increases”.**
*


From this statement, it is frequently concluded that *entropy* always increases, and since time also always increases, there must be a correlation between the two concepts, hence the existence of a “thermodynamic arrow of time”.

However, a close examination of Clausius’ statement as well as those of many others demonstrates that people either ignore or forget that entropy is a *state function*. As such, entropy is defined for any well-defined thermodynamic system at equilibrium. In this sense, entropy cannot be a time-dependent quantity.

The explicit association of the Second Law with the Arrow of Time was first stated by Eddington, in his book ***The Nature of the Physical World*** (1928) [1].

Quotations from Eddington feature in most popular science books, as well as in some textbooks on thermodynamics. Here are two relevant quotations from Eddington’s (1928) book [1]. The first concerns the role of entropy and the Second Law, and the second introduces the idea of “time’s arrow”.


*1. “The practical measure of the random element which can increase in the universe but can never decrease is called entropy…*



*“The law that entropy always increases, holds, I think, the supreme position among the laws of Nature”.*



*2. “Let us draw an arrow arbitrarily. If as we follow the arrow we find more and more of the random element in the state of the world, then the arrow is pointing towards the future; if the random element decreases the arrow points towards the past”.*



*“This follows at once if our fundamental contention is admitted that the introduction of randomness is the only thing which cannot be undone. I shall use the phrase ‘time’s arrow’ to express this one-way property of time which has no analogue in space”.*


In the first quotation, Eddington reiterates the unfounded idea that “entropy always increases”. Although it is not explicitly stated, the second quotation alludes to the connection between the Second Law and the Arrow of Time. This is clear from the (erroneous) association of the “random element in the state of the world” with the “arrow pointing towards the future”.

Clearly, Eddington confused the concept of the absolute always with the practical always. This erroneous identification of the apparent irreversibility of some process with the Arrow of Time appears even today in many popular science books.

Reading through Eddington’s entire book *The Nature of the Physical World*, one can find many pompous statements but not a single correct statement about entropy and the Second Law. For more details on Eddington’s book, see Ben-Naim (2020) [2].

Incredible as it may sound, some authors even equate entropy with time. For instance, Scully, in his book *The Demon and the Quantum* (2007) [3] writes:


*“Entropy not only explains the arrow of time; **it also explains its existence**; it is “time”.*


This is, of course, not true; Entropy does not explain the existence of anything!

Actually, Scully (2007) [3] uses the equality sign “=” in order to express the identity of time’s arrow and entropy:


*“The statistical time concept that entropy = time’s arrow has deep and fascinating implication”.*


This statement is meaningless.

Another meaningless statement about the “entropy of the universe” is found in the book by Atkins (2007) [4]:


*“The entropy of the universe increases in the course of any spontaneous change. The keyword here universe, as always in thermodynamics, the system together its surroundings”.*


Unfortunately, this is an empty statement. The entropy of the universe is not defined. It is therefore meaningless to talk about the changes of the entropy in the universe. Yet, most writers on entropy talk about the ever-increasing entropy of the universe and relate the direction of this change to the Arrow of Time. (For more details, see Ben-Naim, 2015, 2018, 2020) [2,5,6].

Entropy is a state function. This means that entropy is defined for a well-defined thermodynamic system at equilibrium. Thus, if you have a glass of water or a bottle of wine at some specific temperature *T*, pressure *P*, and some composition *N* (and neglecting all kinds of external fields such as electric, magnetic, or gravitation), then the entropy of the system is well-defined. It is a function of the specified variables; we write it as $S=S\left(T,P,N\right)$, and we say that the entropy is a state function. This means that to each state of a thermodynamic system, one can assign a value of the entropy. This value belongs to a specific system at a specific state of equilibrium. For some simple systems, e.g., ideal gases, we can calculate the value of the entropy. For some other more complicated systems, we can measure the entropy up to an additive constant, or equivalently, we can measure the changes in entropy for the same system at two different states. We write it as:$$∆S=S\left(T_{2},P_{2},N_{2}\right)-S\left(T_{1},P_{1},N_{1}\right).$$

In this characterization of the system, the entropy is a function of the variable *T*, *P*, *N*. It is not a function of time! It is never a function of time!

There is one special set of thermodynamic variables characterizing an isolated system for which the Second Law may be formulated in terms of entropy. Such a system is characterized by a fixed energy *E*, fixed volume *V*, and fixed composition *N*. Clearly, such a perfect isolated system does not exist. There are always some interactions with external fields (e.g., gravitational) that affect the state of the system. However, as an idealized case, it is a very convenient system, and it is also the system on which the entire edifice of statistical mechanics was erected.

Given the state of an isolated system (*E*, *V*, *N*), the entropy is defined for such a state. Again, the entropy of this system is a function of the variables *E*, *V*, and *N*. It is not a function of time!

The Second Law of Thermodynamics states that if we remove any internal constraints of an isolated system, the ensuing spontaneous process will always cause entropy to either increase or remain unchanged. The simplest example is the removal of the partition separating two compartments, as shown in Figure 1. Once the partition is removed, the system will proceed to a new equilibrium state (shown on the very right-hand side of the figure) with larger entropy. From this fact, people (and perhaps also Clausius himself) concluded, erroneously, that entropy will change continuously from its initial state (before the removal of the partition) to its final state.

Unfortunately, the Second Law does not state that entropy increases with time. It does not state that the entropy of any system increases with time. It does not even state that the entropy of an isolated system increases with time. Entropy is simply not a function of time, hence **there is no thermodynamic Arrow of Time**!

What the Second Law states is that:


*“When we remove a constraint of an isolated system, the system will move to a new equilibrium state, having higher entropy”.*


Thus, ∆S>0 is the difference between the entropies of the same system, between two different states—not between two different times.

There is another aspect of the Second Law in its probability formulation that might suggest that a one-way direction of irreversible processes is related to the ever-increasing time—the so-called Arrow of Time. In the next two sections, we show that while it is true that irreversible processes (in a macroscopic system) can never be reversed spontaneously, the word “never” used here is only in practice, i.e., never with very high probability. Time, on the other hand, can only increase, it can never decrease. Here, the word never is in the absolute sense. Therefore, it follows that the Arrow of Time may not be identified with the one-way direction of irreversible processes.

There are other “theoretical” arguments that seem to point out that entropy increase is always in the same direction as that of the Arrow of Time. The most seemingly “convincing”, argument is Boltzmann’s H-theorem. We shall not discuss this topic in this article. In Ben-Naim (2018) [6], we show that Boltzmann’s H-function is indeed a function of time. Unfortunately, H(t) does not represent the time dependence of entropy, as numerous authors have erroneously concluded [2,6].

## 2. What Is Time?

To the best of my knowledge, there is no definition of time in terms of some other, more primitive terms. The most well-known quotation regarding the non-existence of a definition of time is from St. Augustine and dates back to the fourth century. In his Confessions, St. Augustine asks:


*“What then is time?”*


Then, he facetiously answers:


*“If no one asks me, I know what it is.*



*If I want to explain it to someone who asks me, I do not know”.*


Although this quotation goes back centuries, it is still valid to this day. We all know what time is, but no one knows how to define it.

Some authors of popular science books not only promise to define time but also offer to the readers a precise definition of time. In the prologue of Carroll’s book *From Eternity to Here* (2010) [7], the author promises that:


*“by the time you read the entire book you will get a precise definition of Time”.*


Unfortunately, this promise is not fulfilled. If you read the entire book, you will not find any definition of time, certainly not a “very precise definition” of time.

When Einstein was asked about the definition of time, he answered:


*“When I say that the train arrives here at 7 o’clock, I mean that the pointing of the small hand of my watch to 7, and the arrival of the train are simultaneous events”.*


Thus, according to Einstein, the only practical “definition of time” is simple—the reading on our clocks.

But what does the reading of the clock tell us?

The clock, any clock, counts the number of times some cyclic or periodic phenomenon occurs [8]. The “counting number” could be the number of revolutions of the earth around the sun (which is used to count the number of years passed), or the number of rotations of the earth about its axis (which we use to count the number of days passed), or the number of full swings of a pendulum (as Galileo Galilei used in his experiment on falling bodies). This is true for modern clocks in which we count the number of vibrations of a crystal or the number of wavelengths of an electromagnetic wave. Thus, the most important aspect of the time is this:

Time, as a counting number, can only increase; it can never decrease.

In order to appreciate the meaning of this sentence, consider the following everyday counting number:

The number of books ever written can never decrease. Even if an entire library is burned, and all its books turn into ashes, the counting of the “number of books ever written” does not decrease. This number might cease to increase, but as long as people (or perhaps also robots or computers) continue to write books, this number will increase.

Let us go back to counting the units of time. First, when we count the number of days, the number of hours, or the number of ticks on our clocks, we count the number of repeated periodic or cyclic phenomena. We assume that each period took the same time. In the counting of time of any clock, we count the number of cycles, and we assume that each cycle took the same time.

The most important aspect of the counting number is it can never decrease. Here, the word never is an absolute never. Since time is measured by a counting number, it follows that time can never decrease—again, “never” in an absolute sense.

In the next section, we will discuss an example of an irreversible process, which also “never” occurs in a reversed direction. This never is in a practical sense, not in an absolute sense.

In addition, we believe that the time-counting number will always increase—always in an absolute sense. Of course, one might argue that my counting, or your counting, might come to an end when no one will be around to do the counting, or when no clocks exist. In a world where no cyclic processes exist, there is nothing to count, and no one to count, and perhaps time will lose its meaning. However, as long as there exists some cyclic process, I believe that this counting will continue to increase, independent of anyone who will perform the counting.

To summarize: When we say that the clock measures the time, we actually mean that it counts the number of periodic or cyclic processes. Thus, the most important property of time is that it behaves as a counting number, implying that it will “never” decrease. For more details, see Ben-Naim (2021) [8].

## 3. Irreversible Processes, Time, and the Second Law

In this section, we introduce two formulations of the Second Law: the thermodynamic and the probability formulation. In my opinion, the latter is a more general and more useful formulation of the Second Law. There is a great amount of confusion regarding these two different formulations. For instance, one of the thermodynamic formulations, employed by Callen (1985) [9], is:

When we remove a constraint from a constrained equilibrium state of an isolated system, the entropy can only increase; it will never decrease.

The “never” is in an absolute sense. On the other hand, the probability formulation, as we shall discuss below, is of a statistical nature; the system will never return to its initial state. Here, “never” is not absolute but “in practice”.

It was Clausius who first formulated the Second Law. Basically, Clausius observed, as every one of us does, that there are many processes that occur in nature spontaneously and always in one direction. Examples abound; see Figure 2.

I. Take gas in a small box and open it within a larger empty box. You will *always* observe that the gas will expand and fill the entire volume of the bigger box.

II. Take two gases, say argon and neon, in different compartments separated by a partition. Remove the partition, and you will *always* observe a spontaneous mixing of the two gases.

III. Take two pieces of iron, one at 300 ℃ and the second at 100 ℃. Bring them to thermal contact. You will *always* observe a spontaneous flow of heat from the body with the higher temperature to the body with the lower temperature. The hotter body will cool down; the cooler body will heat up. At equilibrium, the two bodies will have a uniform temperature. If the two bodies have the same mass, the final uniform temperature will be 200 ℃.

Why do we see the process going in one direction? The fact that we observe such a one-way or one-directional process has led many to associate the so-called Arrow of Time with the Second Law, more specifically with the “tendency of entropy to increase”.

Unfortunately, all the processes we observe here are one-directional, or irreversible, only in practice, and not in an absolute sense.

Before we continue, it must be said that the entropy changes for all these processes are positive when the entire systems are isolated. One can reverse all of these processes by using external agents.

As you must have noticed, I have italicized the word “always” in the description of the three processes shown in Figure 2. Indeed, in all of these examples, you never observe the reverse process spontaneously; the gas *never* condenses into a smaller region in space, the two gases *never* un-mix spontaneously after being mixed, and heat *never* flows from a cold to a hot body. Note again that I italicized the word “never” in the previous sentences. Indeed, we never observe any of these processes occurring spontaneously in the reverse direction. For this reason, the processes shown in Figure 2 (as well as many others) are said to be irreversible. The idea of the absolute irreversibility of these processes was used to identify the direction of these processes with the so-called Arrow of Time. This idea is not only not true; it is also erroneously associated with the very definition of entropy. One should be careful with the use of the words “reversible” and “irreversible” in connection with the Second Law. There are several, very different, definitions assigned to these words. For more details, see Ben-Naim (2011, 2015, 2016, 2019) [5,10,11,12]. Here, we point out two possible definitions of the term irreversible.

1. We never observe that the final state of any of the processes in Figure 2 returns to the initial state (on the left-hand side of Figure 2) spontaneously.

2. We never observe the final state of any of the processes in Figure 2 returning to the initial state and staying in that state.

In case 1, the word never is used in “practice”. The system can go from the final to the initial state. In this case, we can say that the initial state will be revisited. However, such a reversal of the process would occur once in many ages of the universe. Therefore, this is practically an irreversible process; we will “never” observe such a reversal in practice.

In case 2, the word never is used in an absolute sense. The system will never go back to its initial state and stay there! In thermodynamic terms, the system’s entropy will never decrease spontaneously.

Initially, Clausius formulated a “restricted” Second Law, namely that heat does not flow spontaneously from a cold to a hot body. However, he later postulated that there exists a quantity, which he called entropy, that is assigned to any macroscopic system and that always increases when a spontaneous process occurs. This was the birth of the Second Law of Thermodynamics.

Soon after Clausius formulated the Second Law, scientists proved that various particular formulations were all equivalent to each other. The proof of the equivalency appears in any textbook of thermodynamics. Today, we can calculate the change in entropy, and we find that whenever a spontaneous process occurs in an isolated system, the entropy of the system always increases or remains unchanged. We must emphasize that by “entropy changes” we mean the difference in entropy between two values of the entropy of a system at two different equilibrium states.

Notwithstanding the enormous success and the generality of the Second Law, Clausius made one further generalization of the Second Law.


**The entropy of the universe always increases.**


This formulation is an unwarranted over-generalization. Neither Clausius’ definition, nor any other definition of entropy, is applicable to the entire universe. It is unfortunate that the meaningless concept of the “entropy of the universe” appears quite often in recent popular science books.

The entropy formulation of the Second Law applies only to isolated systems. We shall formulate it for a one-component system having *N* particles. If there are *k* components, then *N* is reinterpreted as a vector comprising the numbers $\left(N_{1},N_{2},\ldots ,N_{k}\right)$ where $N_{i}$ is the number of particles of species *i*.

***The entropy of an unconstrained isolated system*** 
$\left(E,V,N\right)$
***, at equilibrium is larger than the entropy of any possible constrained equilibrium states of the same system.***


Note that this formulation uses only macroscopic quantities. Also, it applies only to equilibrium states. The entropy formulation means that if we remove any of the constraints in any possible constrained equilibrium system, the entropy will either increase or remain unchanged.

Therefore, an equivalent formulation of the Second Law is:


**
*Removing any constraint from a constrained equilibrium state of an isolated system will result in increasing (or no change) entropy.*
**


In many textbooks, as well as in popular science books, you might find “formulations” of the Second Law as:


**
*The entropy tends to increase.*
**



**
*The entropy of the universe always increases.*
**



**
*The entropy of an isolated system increases until it reaches a maximum.*
**


All of the above sound similar, but in fact, they are different, and all are wrong. The entropy, by itself, does not tend to increase! The entropy of the universe is not defined! The entropy of an isolated system is defined for an equilibrium state. As such, it does not “increase until it reaches a maximum!”

We also quote here the relationship between the change in entropy and the ratio of the probabilities:(1)$$\frac{Pr\left(final\right)}{Pr\left(initial\right)}=\exp\left[\left[S\left(final\right)-S\left(initial\right)\right]/k\right]$$

One should be very careful in interpreting this equality. The entropy change in this equation refers to the change from state (a) to state (c) in Figure 1, and both are equilibrium states. The probability ratio on the left-hand side of the equation is for the states (b) and (c). The probability of state (a) and of state (c) is one! The reason is that both state (a) and state (c) are equilibrium states.

There are other thermodynamic formulations of the Second Law. The entropy formulation is valid for an isolated system, i.e., (*E*, *V*, *N*) system. The Helmholtz formulation is valid for a (*T*, *V*, *N*) system, and the Gibbs formulation is valid for a (*T*, *P*, *N*) system. Most people who apply the Second Law forget these restrictions and take the liberty of applying, say, the entropy formulation of the Second Law to any system, including living systems.

We shall now quote three connections between the probability ratio and the difference in a thermodynamic quantity.

The ratio of the probabilities in the initial and final states is related to the entropy change in an isolated system, to the Helmholtz energy change in a *T*, *V*, *N* system, and to the Gibbs energy change in a *T*, *P*, *N* system.

We present here these three equations:(2)$$\frac{Pr\left(final\right)}{Pr\left(initial\right)}=\exp\left[\left[S\left(final\right)-S\left(initial\right)\right]/kT\right]\;\frac{Pr\left(final\right)}{Pr\left(initial\right)}=\exp\left[-[A\left(final\right)-A\left(initial\right)]/kT\right]\;\frac{\mathrm{Pr}\left(final\right)}{\mathrm{Pr}\left(initial\right)}=\exp\left[-[G\left(final\right)-G\left(initial\right)]/kT\right]$$

The first is valid for an $\left(E,V,N\right)$ system, the second is valid for a $\left(T,V,N\right)$ system, and the third for a $\left(T,P,N\right)$ system. The first equation reduces to Boltzmann’s formulation when all the microscopic states have equal probabilities, in which case the probability ratio is equal to the ratio: *W*(*final*)/*W(initial).* Note that sometimes, *W* itself is equated to the probability (Pr) and not the ratio. This is not true since the probability is a number between zero and one, whereas *W* could be any number.

Note that the probability ratio is the same in all the equations (2). It is therefore clear from these equations that the probability formulation of the Second Law, which will be stated below, is far more general than any of the thermodynamic formulations in terms of either entropy, Helmholtz energy, or Gibbs energy.

Let us state the probability formulation of the Second Law for this particular example.

We start with an initial constrained equilibrium state. We remove the constraint, and the system’s configuration will evolve, with a probability of (very nearly) one, to a new equilibrium state, and we shall never observe a reversal to the initial state. “Never”, here, means never in our lifetime, and never in the lifetime of the universe. It is not “never” in an absolute sense.

This formulation is valid for large N. It is also valid for any initial constrained equilibrium state. In the next section, we show a few simulated results for the expansion process with different values of *N.*

## 4. Some Simulated Processes of Expansions

In the following, we will present a few processes of the expansion of an ideal gas with different numbers of particles. In all these processes, we remove a partition between the two compartments, and the particles will move to occupy the larger volume spontaneously. However, from time to time, we will observe all the particles in one compartment (either the left or the right one). Once all the particles are in one compartment (and this will occur with decreasing probability as the number of particles increases), the particles will expand again to occupy all the available volume. In order to stay in one compartment, the partition should be replaced at its original position. Of course, this will never (in an absolute sense) occur spontaneously. For more details on this topic, see Ben-Naim (2020) [2].

Consider a system of *N* non-interacting particles (ideal gas) in a volume 2V at constant energy *E*. We divide the system into two compartments, L and R, each of volume *V*, as in Figure 1.

In the following simulations, we start with all particles (*N*) in the left compartment, L, and follow the number of particles in this compartment, denoted *n,* as a function of time.

***The case of two particles:*** N=2

Suppose we have the total of N=2 particles (Figure 1). In this case, we have the following possible values for *n*:(3)$$n=0\;n=1\;n=2\;,\;P_{N}\left(0\right)=\frac{1}{4},\;P_{N}\left(1\right)=\frac{1}{2},\;P_{N}\left(2\right)=\frac{1}{4}$$

This means that on average, we can expect to find the configuration *n* = 1 (i.e., one particle in each compartment) about half of the time, but each of the configurations (n=0 and n=2) only a quarter of the time (Figure 3a).

If we start with all the particles (*N* = 2) in the left compartment, we shall find that the system will “expand” from *V* to 2*V*. However, once in a while, the two particles will be found in one compartment.

Figure 3a shows some simulated results for this experiment. We start with all the particles in the left compartment (L). We then choose a particle at random and transfer it to a random compartment, in this case to either L or R. (More details on the program of the simulations with different numbers of particles and different numbers of cells may be found in Ben-Naim (2010)) [2,13].

We see that the number of particles in L starts with two, and then the number fluctuates between zero, one, and two. In this particular run, we found about four snapshots in which there were two in L and zero in R; about six with zero in L and two in L; and ten with one in L and one in R. Clearly, in this particular case, there is no indication that the process is “irreversible” or that there is a one-directional evolution of the state of the system.

***The case of four particles:*** N=4

The case of N=4 is shown in Figure 3b. Here, we see fewer visits to the initial state as well as to the state with zero particles in L. Most of the states are such that there are two particles in each compartment. Again, there is no clear “one-directional” evolution of the state of the system. The system does return to the initial state.

***The case of four particles:*** N=8

The case of N=8 is shown in Figure 3c. Here, we do not see visits to the initial state. However, there is no clear “one-directional” evolution of the state of the system. If we wait longer, the system will visit the initial state.

***The case of 256 particles:*** N=256

For $2^{8}=256$ particles, we see that the number of particles in L steadily, almost monotonically, decreases from the initial value of 256 to about 256/2 = 128, and then fluctuates about this number (Figure 4a).

***The case of 1024 particles:*** N=1024

Finally, we show in Figure 4b the results for the case of $N=2^{10}=1024$. Again, we notice that the curves became smoother, nearly monotonic. While it is clear that in this case no return to the initial state is observed, we cannot conclude that the process is strictly irreversible. If we wait long enough, we shall observe visits to the initial state—such events are not impossible but are extremely improbable. They could occur in about 1 in 2^1024^ snapshots, which is a huge number—about 10^300^, i.e., 10 followed by 300 zeros!

One can imagine that for *N* of the order of 10^23^, the curves we have seen in Figure 3 and Figure 4 will become almost monotonic, and once we reach the equilibrium value, it will stay there forever. This “forever” does not mean that the process is strictly irreversible; it only means that reversing to the initial state can occur in about $2^{10^{23}}$ snapshots. Such a number is unimaginable; we shall not see such a reversal, not in our lifetime, and not in the lifetime of the whole universe.

We can conclude that for $N\approx 10^{23}$, the number *n* will change monotonically with time, and once it reaches the value of *N/2*, it will stay there “forever”. This “forever” is not an absolute forever, but a practical forever. Thus, the concept of irreversibility that we observe is merely an illusion. It is a result of our relatively short lifetime.

The abovementioned specific example of expansion provides an explanation for the fact that the system will “always” evolve in one direction and “always” stays at the equilibrium state once that state is reached. The tendency towards a state of larger probability is equivalent to the statement that events that are supposed to occur more frequently will occur more frequently. This is plain common sense. The fact that we do not observe deviations from either the monotonic climbing of *n* towards $n^{*}$ or staying close to $n^{*}$ is a result of our inability to detect small changes in *n*.

## 5. Summary and Conclusions

What is the entropy change in the process of expansion, as seen in Figure 1? In this process, the macrostate of the system is defined initially by $\left(E,V,N\right)$. The corresponding value of the entropy is $S\left(E,V,N\right)$. The final macrostate is characterized by $\;\left(E,2V,N\right)$, and the corresponding value of the entropy is $S\left(E,2V,N\right)$. In between the two macrostates $\left(E,V,N\right)$ and $\left(E,2V,N\right),$ the macrostate of the system is not well-defined. A few, intermediate states are shown in Figure 1. While *E* and *N* are the same as in the initial state, the “volume” during the expansion process of the gas is not well-defined. It becomes well-defined only when the system reaches an equilibrium state. Therefore, since the volume of the system is not well-defined when the gas expands, the entropy is also not well-defined. We can say that the entropy changes abruptly from $S\left(E,V,N\right)$ to $S\left(E,2V,N\right)$, and that this change occurred at the moment the system reached a final equilibrium state.

One can also adopt the point of view that when we remove the partition between the two compartments, the volume of the gas changes abruptly from *V* to 2*V*. Although the gas is initially still in one compartment, the total volume accessible to all particles is 2*V*. If we adopt this view, then at the moment we removed the partition, the volume changed from *V* to 2*V*, and the corresponding change in entropy is $S\left(E,2V,N\right)-S\left(E,V,N\right)$. This change occurs abruptly at the moment we remove the partition. Initially, it has the value $S\left(E,V,N\right)$ before the removal of the partition, and it reaches the value of $S\left(E,2V,N\right)$ when the system reaches the new, final equilibrium state. In all the intermediate states, the entropy is not defined.

It should be noted, however, that we could devise another expansion (referred to as quasi-static) process by gradually moving the partition between the two compartments. In this process, the system proceeds through a series of equilibrium states, and therefore the entropy is well-defined at each of the points along the path leading from $\left(E,V,N\right)$ to $\left(E,2V,N\right)$. In this process, the entropy of the gas will gradually change from $S\left(E,V,N\right)$ to $S\left(E,2V,N\right)$.

Note that the sequences of states in the spontaneous process are different from the quasi-static process. In the latter, the states as well as the entropy of the gas are well-defined along the entire path from the initial to the final equilibrium states, whereas in the spontaneous expansion, neither the states nor the entropy are defined along the path leading from the initial to the final state.

Thus, we can conclude that the apparent irreversibility of all the processes we deemed to be “irreversible” is only an illusion. We shall never observe any significant deviation from this new equilibrium state, not in our lifetime, and not in the universe’s lifetime, which is estimated to be about 15 billion years, i.e.,
$$\frac{\mathrm{Pr}\left(final\;configuration\right)}{\mathrm{Pr}\left(initial\;configuration\right)}\approx infinity$$

This is the essence of the probability formulation of the Second Law for this particular experiment. This law states that by starting with an equilibrium state where all particles are in L and removing the constraint (the partition), the system will evolve to a new equilibrium configuration that has a probability overwhelmingly larger than the initial configuration.

The distinction between the strictly mathematical monotonic change and the practical change is important. The process is mathematically always reversible, i.e., the initial state will be revisited. However, in practice, the process is irreversible; we shall never see the reversal to the initial state.

## Figures and Tables

**Figure 1 entropy-25-01297-f001:**
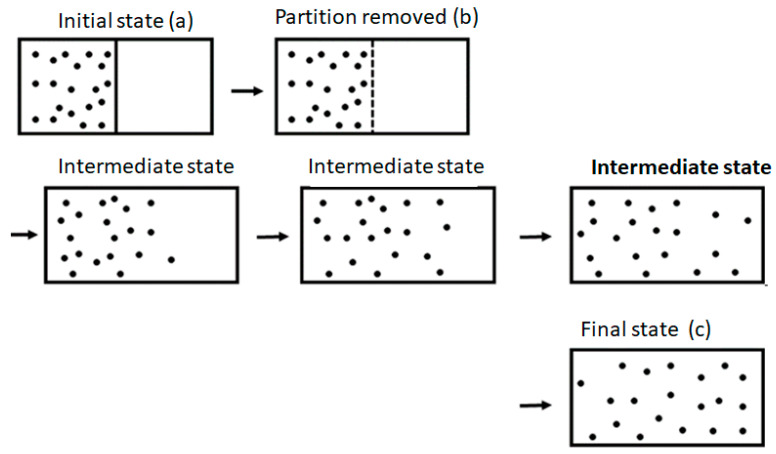
The initial equilibrium state (**a**). The state right after we removed the partition (**b**). The final equilibrium state (**c**). A few intermediate states in the expansion process.

**Figure 2 entropy-25-01297-f002:**
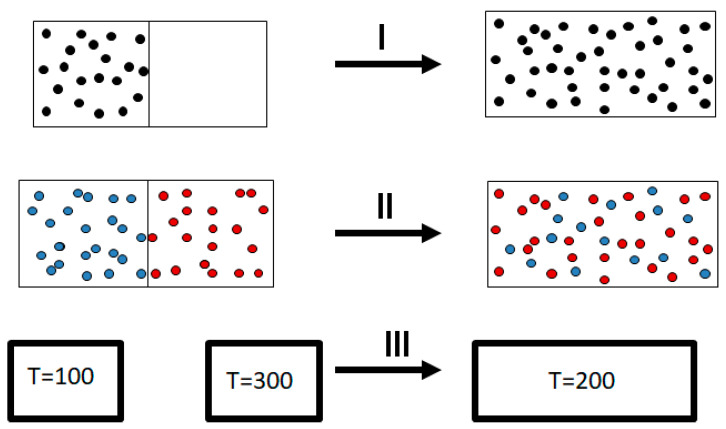
Three irreversible processes: I. Expansion of an ideal gas. II. Mixing of ideal gases. III. Heat transfer from a hot to a cold body.

**Figure 3 entropy-25-01297-f003:**
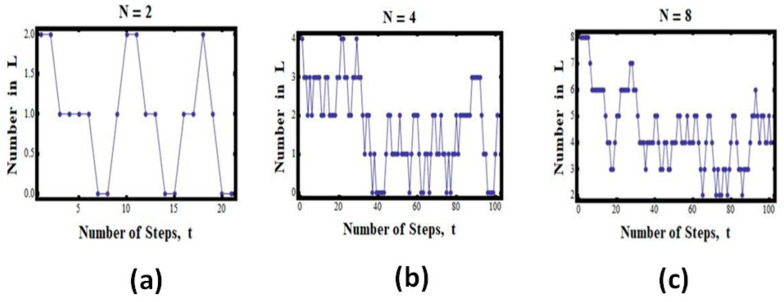
Simulated results for *N* = 2 (**a**), 4 (**b**), and 8 (**c**). The number of particles (*n*) in the left compartment as a function of the number of steps in the simulated experiment.

**Figure 4 entropy-25-01297-f004:**
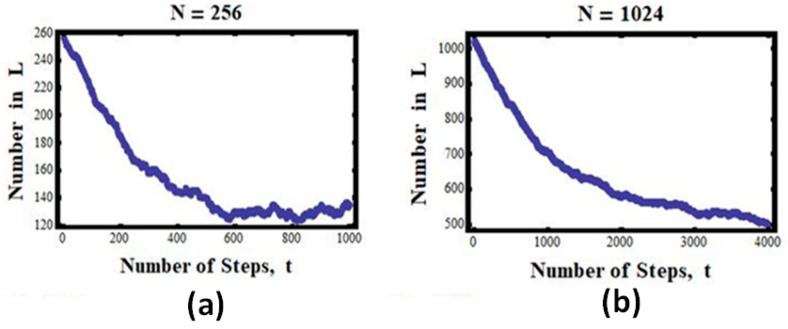
Simulated results for *N* = 256 (**a**) and 1024 (**b**). The number of particles (*n*) in the left compartment as a function of the number of steps in the simulated experiment.

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
