# Peer review of "Is Time’s Asymmetry Related to Irreversible Processes and the Second Law?"

_entropy, 2023, doi:10.3390/e25091297_

Round 1
Reviewer 1 Report
The paper "Is time’s asymmetry related to irreversible processes and the Second Law?" by Arieh Ben-Naim is concerned with explaining why the second law of thermodynamics has nothing to do with time or with time's arrow. Both conclusions are, in spite of a large amount of literature that argues to the contrary, correct. It is therefore very important to communicate this true fact clearly, so it does not get forgotten in an overwhelming background of incorrect chatter. I am therefore very sympathetic to the message of the paper and think it must be communicated.
Let me summarise, what I find to be the features of the paper:
A large part of the author's argument uses the correct statement that the concept of entropy is not an unambiguous concept outside of equilibrium thermodynamics of confined systems. It is therefore not clear "which entropy" should be meant by the statement: "The entropy of the universe always increases." I agree that such a blanket statement is empty unless the terms are well defined. Besides this, the paper makes additional points that I also agree with:
1. Entropy, for systems, for which it is actually well defined, increases f.a.p.p. only, while time, when correctly defined as a counting measure increases unconditionally.
2. The entropy of the universe is not even defined, even if the universe were an equilibrium thermodynamic system.
Moreover, the author includes a section with simulations of a simple equilibration process to illustrate the f.a.p.p. nature. This simulation is very elementary and no technical details about nature are stated, but the example suffices for educational purposes.
All the positive facts that I stated about the paper do however not overcome the important fact that the paper is not reporting new results. The paper is therefore very well suited as an educational paper, but I do not classify it as a research paper. This educational, rather than research nature of the paper is mirrored by the fact that the paper does not cite any research article and most references are references to popular works by the author.
I therefore recommend the publication of the paper in a journal that is concerned with the education of physics.
Please correct the typo on p. 7 l. 307: "We shall quote thee ..." -> "... the ..."
Author Response
Respond to reviewer number 1:
Thanks for the comments. I am glad you agree with my statements.
I agree that this paper does not report on new result. From the very beginning it was agreed with the editors of Entropy that the article is not intended to be a research article, but rather a kind of personal review of what I have already published.
I corrected the sentence on line 307.
thanks.
Reviewer 2 Report
Generally, a very clearly written article, which will be useful for a general readership.The reported simulations ($4) provide an excellent illustration.
The manuscript mentions several times that the concept of the 'entropy of the universe' is meaningless. Perhaps it is an unfortunate use of this expression but Penrose (amongst others) employs this concept quite frequently. A brief paragraph on why Penrose should not use this phrase would be appropriate.
Throughout the article references appear without page numbers. This should be corrected.
In $2 (p. 4) clock time (Einstein) is said to to based on regular cyclic events but they must also be invariant. And is it really true that 'we assume that each cycle took the same time' (p. 4), since triangulation method can be used to support this assumption? In other words, this assumption can be tested.
Line 307: should read 'three connections'
Line 353: should read 'for' n
Line 408: should read: an explanation 'for'...
Author Response
Respond to reviewer number 2:
There are many authors who discussed the “entropy of the universe.”
I have criticized many of these authors, including Penrose, in my book published in (2020). I refrained from doing so in this article.
I do not think I should quote page numbers, since most of my criticism is about the whole book cited, not a particular line or page.
I agree with the comment about the “this assumption can be tested”
In fact, I have discussed that in reference 8. However, as I showed in reference 8, even after testing this assumption with a finer clock, there remains the doubt about the
‘size of the cycle’ for the finer clock. This doubt remain even after having the most accurate clock.
I corrected all three typos in the revised manuscript.
Thanks.
Round 2
Reviewer 1 Report
The manuscript can be published, in particular in lingth of the explanations given by the author on the purpose of the paper and the minor corrections to the draft.
Reviewer 2 Report
In my opinion you should add page numbers to help the reader locate the quote and see it in context.